# Phenotypic Heterogeneity among *GBA* p.R202X Carriers in Lewy Body Spectrum Disorders

**DOI:** 10.3390/biomedicines10010160

**Published:** 2022-01-12

**Authors:** Valerio Napolioni, Carolyn A. Fredericks, Yongha Kim, Divya Channappa, Raiyan R. Khan, Lily H. Kim, Faria Zafar, Julien Couthouis, Guido A. Davidzon, Elizabeth C. Mormino, Aaron D. Gitler, Thomas J. Montine, Birgitt Schüle, Michael D. Greicius

**Affiliations:** 1Department of Neurology and Neurological Sciences, Stanford University School of Medicine, Stanford, CA 94305, USA; carolyn.fredericks@yale.edu (C.A.F.); yonghakimis@gmail.com (Y.K.); rrk2147@columbia.edu (R.R.K.); lilyhkim@stanford.edu (L.H.K.); bmormino@stanford.edu (E.C.M.); greicius@stanford.edu (M.D.G.); 2Department of Pathology, Stanford University School of Medicine, Stanford, CA 94305, USA; divyac2@stanford.edu (D.C.); fzafar@stanford.edu (F.Z.); tmontine@stanford.edu (T.J.M.); bschuele@stanford.edu (B.S.); 3Department of Genetics, Stanford University School of Medicine, Stanford, CA 94305, USA; jcouthouis@ultragenyx.com (J.C.); agitler@stanford.edu (A.D.G.); 4Department of Radiology, Stanford University School of Medicine, Stanford, CA 94305, USA; gdavidzon@stanford.edu

**Keywords:** Gaucher’s disease, glucocerebrosidase, genetics, Lewy body dementia, mutation, neuropathology, Parkinson’s disease, sequencing

## Abstract

We describe the clinical and neuropathologic features of patients with Lewy body spectrum disorder (LBSD) carrying a nonsense variant, c.604C>T; p.R202X, in the glucocerebrosidase 1 (*GBA*) gene. While this *GBA* variant is causative for Gaucher’s disease, the pathogenic role of this mutation in LBSD is unclear. Detailed neuropathologic evaluation was performed for one index case and a structured literature review of other *GBA* p.R202X carriers was conducted. Through the systematic literature search, we identified three additional reported subjects carrying the same *GBA* mutation, including one Parkinson’s disease (PD) patient with early disease onset, one case with neuropathologically-verified LBSD, and one unaffected relative of a Gaucher’s disease patient. Among the affected subjects carrying the *GBA* p.R202X, all males were diagnosed with Lewy body dementia, while the two females presented as PD. The clinical penetrance of *GBA* p.R202X in LBSD patients and families argues strongly for a pathogenic role for this variant, although presenting with a striking phenotypic heterogeneity of clinical and pathological features.

## 1. Introduction

Intracellular aggregation of α-synuclein is a pathological hallmark of Lewy body spectrum disorders (LBSD), a heterogeneous group of neurodegenerative diseases that includes Parkinson’s disease (PD), Parkinson’s disease with dementia (PDD), and dementia with Lewy bodies (DLB). Clinical manifestations of LBSD are highly diverse, with high variability in age-at-onset, disease progression, occurrence of motor symptoms, and associated cognitive impairment [1]. Genetic variants in the apolipoprotein E (*APOE*), tau (*MAPT*), α-synuclein (*SNCA*), or glucocerebrosidase (*GBA*) genes have been associated with LBSD [1].

*GBA* variants that are causative for Gaucher’s disease (GD, OMIM#230800) are a significant risk factor for PD and related α-synucleinopathies [2,3]. Almost 500 different *GBA* variants have been associated with GD. Predominantly, recurrent or founder mutations in the *GBA* gene are present in PD patients with Ashkenazi Jewish ancestry (minor allele frequency (MAF) ranging from 10 to 31%). In contrast, *GBA* variants in non-Ashkenazi backgrounds occur at a lower MAF (ranging from 3 to 12%) and encompass a wider array of variants than the handful of founder mutations seen with Ashkenazi ancestry [4,5].

In the current work, we present detailed clinical and pathological case descriptions of two non-Ashkenazi, European-American families with LBSD who carry a rare *GBA* nonsense variant (NM_000157.3:c.604C>T, NP_000148.2:p.R202X, rs1009850780). We present other *GBA* p.R202X carriers based on a structured literature review and database search. We compare the present cases to the previously described cases and describe a pathogenic role for this mutation. Finally, considering the rapid growth of whole-exome and whole-genome sequencing in the clinical research setting, we address challenges related to identifying and interpreting *GBA* variants and considerations for clinical practice and genetic counseling.

## 2. Materials and Methods

### 2.1. Standard Protocol Approvals, Registrations, and Patient Consents

All individuals provided written informed consent before participating in this study for clinical evaluation and genetic analysis. The Stanford Institutional Review Board approved all study procedures.

### 2.2. Genetic Analysis

DNA of living subjects was collected from saliva samples, dried blood spots, or whole blood, while the DNA of the index case from Family 1 was extracted from brain-autopsy tissue.

#### 2.2.1. Family 1

The whole exome was captured using SureSelect Human All Exon (Agilent, Santa Clara, CA, USA.) and sequenced on the Illumina HiSeq2000 platform using a 2 × 100 bp Paired-End chemistry. Raw data were processed using SeqMule [6]. Variants present in the father–son pair, which the two healthy aunts did not carry, were kept, and filtered against 4449 control subjects aged 80 and older (1802 males, 2647 females) from the Alzheimer’s Disease Sequencing Project (ADSP) [7]. Variants not found in any controls were annotated using the Ensembl Variant Effect Predictor toolkit [8]. Sanger sequencing of *GBA* exon 6 was performed using the long-range PCR protocol, as previously described [9]. This protocol allows the specific amplification of the *GBA* gene, avoiding the possibility of detecting false-positive variants due to the existence of the highly homologous (96%) pseudogene (*GBAP1*) located downstream from the *GBA* gene [10].

The index case’s great-granddaughter and her spouse, along with their children, were tested at PerkinElmer Genomics Laboratory using an ad hoc *GBA* Gene Sequencing and Deletion/Duplication Analysis assay. Briefly, genomic DNA extracted from dried blood spots underwent long-range PCR, designed to avoid *GBAP* pseudogene contamination. A custom Agilent SureSelect enrichment kit was used to enrich the regions of interest, and NGS was performed on an Illumina system with a 2 × 100 Paired-End chemistry. NGS results were confirmed by Sanger sequence analysis.

#### 2.2.2. Family 2

Targeted sequencing of PD-related genes was conducted for the second family. Samples were tested using the PD extended NGS panel (Centogene, Rostock, Germany), which covers the entire coding region of the *ADCY5*, *ANO3*, *ATP9A*, *COX20*, *PARK7*, *GBA*, *GCH1*, *GNAL*, *GNE*, *KMT2B*, *LRRK2*, *MCOLN1*, *PRKN*, *PDE8B*, *PDGFB*, *PDGFRB*, *PINK1*, *PLA2G6*, *POLG*, *PRKRA*, *RAB12*, *RAB39B*, *SGCE*, *SLC20A2*, *SNCA*, *THAP1*, *TOR1A*, *VAC14*, *VPS13C, VPS35*, *XPR1,* and 3 *XDP* variants (DSC3, DSC12, and rs41438158) genes, including 10 bp of flanking intronic sequences. Raw sequence data analysis, including base calling, demultiplexing, alignment to the hg19 reference genome, and variant calling was performed using validated in-house software (Centogene, Rostock, Germany). All variants, except benign or likely benign variants, were reported.

### 2.3. Literature and Database Search

A comprehensive literature search was carried out on PubMed using the keywords: “*GBA*”, ‘‘glucocerebrosidase”, ‘‘mutation”, ‘‘variant”, ‘‘dementia”, ‘‘Parkinson” and ‘‘Lewy-body”, applying the following algorithm: (*GBA* OR glucocerebrosidase) AND (dementia OR Parkinson OR Lewy-body) AND (mutation OR variant). Several publicly available genomic databases, including the Genome Aggregation Database (gnomAD) [11], the Greater Middle East (GME) Variome Project [12], and HEX (Healthy Exomes) [13], were queried to determine the frequency of *GBA* p.R202X and its ethnic stratification.

## 3. Results

### 3.1. Case Presentation

#### 3.1.1. Family 1

The index case presented at age 73 to the Stanford Center for Memory Disorders with a three-year history of cognitive decline and parkinsonism. His wife first became concerned about him when he seemed not to understand the sequence of steps for carving a turkey at Thanksgiving. At the same time, she noted motor slowing and a shuffling in his gait. He had a history of dream enactment, an indicator for REM behavior disorder, and subsequently developed visual hallucinations, meeting the criteria for probable DLB. He died at age 74, and a brain autopsy demonstrated Lewy bodies and Lewy neurites in the midbrain, pons, and nucleus basalis of Meynert, as well as a loss of pigmented neurons in the substantia nigra, consistent with a pathologic diagnosis of diffuse Lewy body disease (Braak 6/6, NIA-AA Alzheimer’s disease neuropathologic change 3/3) (Figure 1). Neurofibrillary tangles were confined to the hippocampus, with the presence of mild arteriolosclerosis. There were no signs of hippocampal sclerosis or TDP-43 inclusions.

A few years after his death, his son, then aged 60, presented to the Stanford Center for Memory Disorders, reporting a two-year history of visual hallucinations and a slow, shuffling gait, with a recent onset of dream enactment. Though he described mild difficulties with executive function, word-finding difficulties, and being less comfortable with driving, he was functionally unimpaired at the time of examination. His symptoms continued to progress slowly, so that at age 62, he was still driving locally, but dressing himself had become difficult due to bradykinesia. He made occasional errors in managing his finances and medications. He enrolled in the Stanford Alzheimer’s Disease Research Center (ADRC) where he underwent research amyloid and tau PET scans. The amyloid scan was read as negative, and the tau scan showed limited tracer uptake in the medial temporal lobes that was consistent with healthy controls at this age. Quantitative PET data for this patient are shown in Figure 2.

These two individuals belong to a large Irish American family with a documented history of dementia and PD across three generations (Figure 3A). The index case had three living sisters (ages 72, 75, and 77) who were contacted to assess their neurologic status. The oldest sister was cognitively healthy following clinical and neuropsychological evaluations at the Stanford ADRC. The youngest sister was presumed unaffected based on a Clinical Dementia Rating [14] score of 0 obtained by a telephone interview. A precise clinical diagnosis for the middle sister was not possible due to a history of developmental delay (presumed secondary to scarlet fever) and confounding severe arthritis symptoms with associated immobility.

Notably, since the initial evaluation of the index case, his granddaughter gave birth to a girl who was diagnosed with neuronopathic-type GD Type III at age 3 (Figure 3A). Genetic testing disclosed the presence of compound heterozygous *GBA* genotype (p.R202X/p.G416S), with the p.G416S mutation (NM_000157.3:c.1246G>A, rs121908311) inherited from the father. Both the mother and the unaffected brother (age 5) were found to be carriers of *GBA* p.R202X (Figure 3A).

#### 3.1.2. Family 2

At 36 years of age, the index case, of British-American ancestry, developed a left arm tremor, reduced left-arm swing, and balance problems. She was diagnosed with early-onset, tremor-predominant PD at age 38. Additional symptoms included anxiety and depression, constipation, and urinary urgency. She was responsive to dopaminergic therapy. She scored a 27/30 (normal range) on the Montreal Cognitive Assessment [15] at age 43. She underwent deep brain stimulation at age 45. Her mother is a carrier of the *GBA* p.R202X variant but did not show symptoms at age 72. Her younger brother has reported tremors, but he has not been formally evaluated. Her maternal grandmother developed unspecified dementia in her late 70s, with family noting that her “conversations went in circles”. In the end, she could neither prepare meals, nor understand dates or holidays. She died at age 86.

### 3.2. Identification of GBA p.R202X Variant

Whole-exome (WES) and targeted sequencing of PD genes in two unrelated European-American families with index cases presenting with LBSD revealed the presence of a rare *GBA* stop-gain variant (NM_000157.3:c.604C>T, NP_000148.2:p.Arg202Ter, rs1009850780, gnomAD [11] v.2.1.1, global MAF = 9.02 × 10^−6^) in all affected subjects, the granddaughter of the index case in Family 1, along with her children, and the unaffected mother of the index case in Family 2 (Figure 3). The variant was not found among the two unaffected sisters of the Family 1 index case, nor in any Alzheimer’s Disease Sequencing Project (ADSP) [7] controls aged 80 and above.

Family 1 members were sequenced using both WES and targeted-sequencing approaches since the diagnosis of GD in the 3-year-old great-granddaughter of the index case prompted independent clinical genetic testing.

WES of the four Family 1 members (the index case and three of his first-degree relatives, including the affected son) yielded 252 million reads for a total of approximatively 25.2 Gbps, resulting in an average depth of target coverage of 68X per sample. The joint variant calling by SeqMule [6] yielded 54,104 variants that were filtered according to their presence in both affected subjects and their absence in the two healthy sisters. We retained only the non-synonymous (missense, frameshift, and nonsense) variants with a minor allele frequency (MAF) < 0.05 that were not present in healthy ADSP [7] controls over 80 years of age. The final list included 32 candidate variants, with the *GBA* p.R202X being the only variant predicted to have a high-impact consequence. A literature search for the other 31 candidate variants did not suggest any possible biological link with the affected subjects’ phenotype. Sanger sequencing confirmed *GBA* p.R202X in the affected son of the index case and its absence from the two unaffected siblings (Figure 4). No DNA was available for *GBA* p.R202X Sanger sequencing of the index case.

Targeted sequencing of PD genes revealed the presence of *GBA* p.R202X in both Family 2 members.

### 3.3. Literature Search of GBA p.R202X Variant Carriers

The systematic literature search for other *GBA* p.R202X carriers, carried out in June 2021, yielded 812 articles published between 1993 and 2021, 190 of which were review articles. We adopted the *GBA* mutation nomenclature according to the Human Genome Variation Society (http://hgvs.org, accessed on 11 October 2021); the old nomenclature was based on amino-acid residue numbering, which excludes the first 39 amino acids of the leader sequence, thus defining the mutation as p.Arg163Ter (p.R163X).

We identified three additional reported subjects carrying the same *GBA* mutation, including one PD patient with early-onset illness beginning at 47 years [16,17,18,19], one case with neuropathologically-verified LBSD [1], and one unaffected relative of a GD patient [20]. Only two non-Finnish European subjects (one Swedish and one Estonian) from gnomAD v.2.1.1 were reported to be heterozygote carriers of *GBA* p.R202X (gnomAD reports the age, between 55 and 60, for one of the two carriers). We did not find additional *GBA* p.R202X carriers in the Greater Middle East (GME) Variome Project [12] or the Healthy Exomes (HEX) database [13].

We describe the clinical and demographic characteristics of the seven *GBA* p.R202X carriers in Table 1.

## 4. Discussion

The presence of the *GBA* p.R202X mutation in both LBSD families argues strongly for a pathogenic role for this variant in LBSD, although with phenotypic heterogeneity. *GBA* p.R202X is a loss-of-function variant that leads to haploinsufficiency, and we would anticipate high penetrance for disease. However, we found profound phenotypic variability among affected family members in the two families, demonstrating rather heterogeneous effects of this *GBA* variant on LBSD. The clinical presentation in the affected *GBA* p.R202X carriers ranged from early-onset PD for the index case of Family 2 (age-at-onset 37 years) to neuropathologically confirmed late-onset DLB for the index case of Family 1 (age-at-onset 70 years). Moreover, the cognitively healthy, 72-year-old mother of the index case in Family 2 also carries the mutation. Such phenotypic heterogeneity provides strong support for the idea that other genetic modifier loci strongly impact the phenotypic presentation of LBSD. For instance, we can speculate that the significant difference in age-at-onset for the two male cases in Family 1 (70 years vs. 58 years) could be attributable to the presence of *APOE**4 allele in the earlier-onset subject. *APOE**4 is now a confirmed risk factor for DLB, independent of any co-morbid Alzheimer’s disease pathology [21]. Nevertheless, we cannot exclude the possibility of other biological factors, not detectable by targeted NGS, as the basis of the phenotypic heterogeneity characterizing the affected members of the two families described here. Indeed, it has been shown that telomere length may influence the occurrence of dementia in PD patients [22]. Additionally, several studies have argued for the existence of genetic modifiers of PD clinical presentation, such as *TMEM106B* [23], *SCNA* [24], and *COMT* [25].

Similarly, sex may play a role. Among the affected subjects carrying the *GBA* p.R202X variant, all males were diagnosed with LBD, while the two affected females presented as PD. This is not entirely surprising, as previous studies have highlighted sex-specific differences in the phenotype of *GBA* mutation carriers. While most men presented with DLB, most women presented with PD [26]; moreover, males with *GBA*-associated PD have a higher burden of trait anxiety and depression than females [27].

Undoubtedly, a deeper biological characterization of the two families described here, obtained by performing a wider genomic investigation (e.g., by whole-genome sequencing and structural variation typing), eventually coupled with the analysis of epigenomic changes (e.g., telomere length and methylation profiling), would help to identify new modifiers of clinical phenotypes in LBSD.

Despite the association of recurrent *GBA* mutations with PD predominantly in the Ashkenazi Jewish population [4], the presented results and literature search both point to a non-Ashkenazi origin for this mutation. However, the *GBA* locus is characterized by frequent recombination events involving both gene conversion and reciprocal recombination [10]. As a result, it is difficult to attribute *GBA* p.R202X to a specific ethnic background. Notably, most studies and clinical tests are based on the genotyping of a limited number of mutations that are most relevant for the Ashkenazi Jewish population. The sequencing of the whole gene will be of paramount relevance in determining *GBA* variants in other ethnic groups that otherwise would be missed.

In this context, special attention must be paid when choosing the genotyping method of *GBA*. Since the coding region of the *GBA* gene is 96% homologous to *GBAP*, long amplicons are generally needed to ensure that only the *GBA* gene is being targeted when genotyping. Moreover, the frequency of *GBA*-*GBAP1* complex rearrangements makes next-generation sequencing (NGS) analysis of *GBA* challenging [28], and adequate strategies (such as *GBA*-specific long-range PCR for library preparation, followed by *GBA*-specific alignment) must be adopted to avoid misdetection of *GBA* recombinant mutations. Because of these challenges, we suspect that the impact of *GBA* mutations on LBSD may have been underestimated in large genetic studies that employed NGS (whole-exome and whole-genome sequencing) and SNP-array-based techniques. Targeted *GBA*-specific sequencing in patients reporting LBSD in the future will help to clarify the prevalence of this highly significant class of pathogenic mutations.

In clinical practice, clinical genetic testing is often performed as a genotyping panel for the most common *GBA* variants (e.g., Invitae GBA carrier screening). In this clinical context, we expect that *GBA* variants that are not commonly present in the Ashkenazi population will be undetected, meaning that clinicians may miss rarer, causal variants in other ethnic backgrounds, including the *GBA* p.R202X described here.

The cases described in the present study represent the first clinical–genetic report on the rare *GBA* p.R202X variant in LBSD. The phenotypic heterogeneity seen in individuals carrying this variant, even within a single family, is striking. Our report also raises the possibility of incomplete penetrance, as the mother of the index case in Family 2 carries the variant but has thus far remained clinically unaffected. These cases emphasize the need for targeted *GBA*-specific sequencing in patients with family histories of LBSD or PD to capture all disease-relevant variants.

## Figures and Tables

**Figure 1 biomedicines-10-00160-f001:**
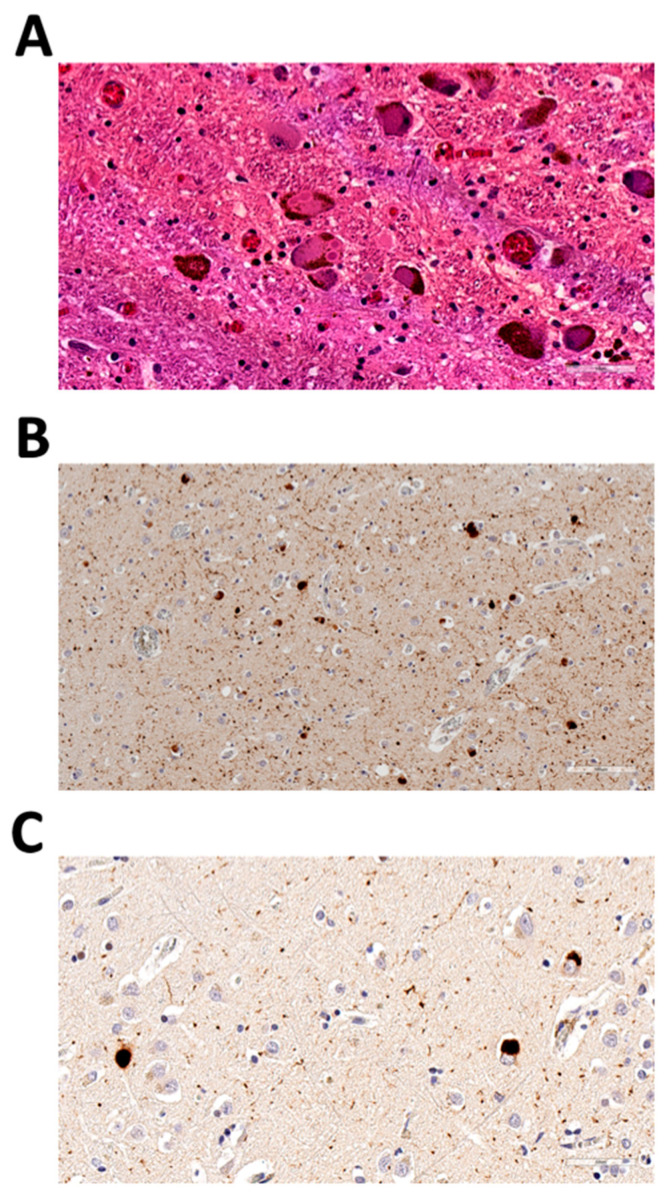
Neuropathologic examination of the index case from Family 1 showing diffuse Lewy body disease. (**A**) Hematoxylin and eosin (HE)-stained section from the locus coeruleus, 40× magnification; (**B**) phosphorylated α-synuclein [pS129]-stained section from the left amygdala, 20× magnification; (**C**) phosphorylated α-synuclein [pS129]-stained section from the right frontal cortex, 40× magnification.

**Figure 2 biomedicines-10-00160-f002:**
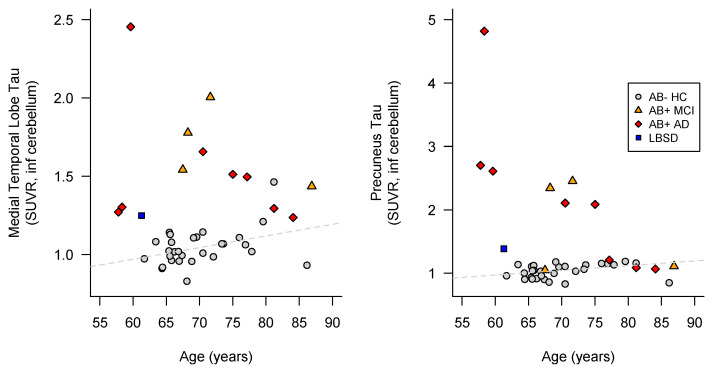
Age and tau PET scans, according to amyloid deposition positivity, in Stanford ADRC participants. Family 1: affected son of the index case is reported as a blue square (LBSD). AB = beta-amyloid; AD = Alzheimer’s disease; HC = healthy control; MCI = mild cognitive impairment.

**Figure 3 biomedicines-10-00160-f003:**
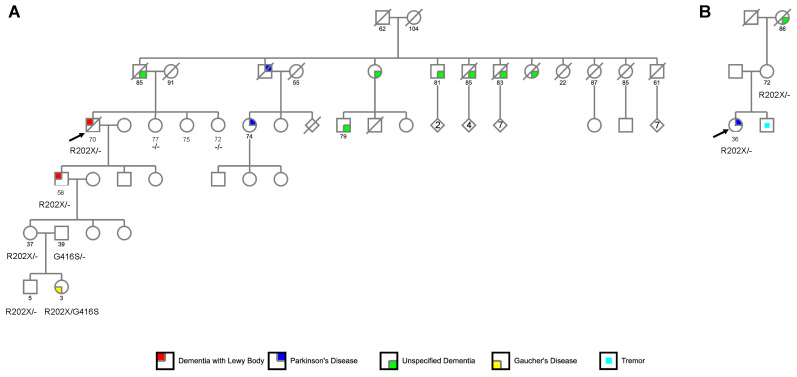
Pedigree of the reported LBSD families. Family 1 (**panel** (**A**)) and Family 2 (**panel** (**B**)). Arrows indicate the index cases.

**Figure 4 biomedicines-10-00160-f004:**
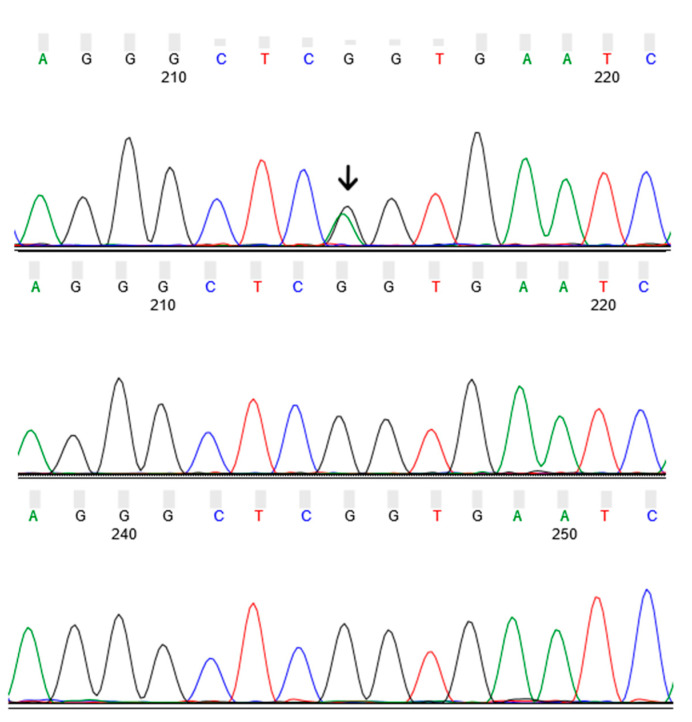
Sanger sequencing results of *GBA* p.R202X in Family 1. The top panel shows the presence of the G/A (heterozygote) genotype in the affected son of the index case and its absence from the two unaffected sisters. No DNA was available for *GBA* p.R202X Sanger sequencing of the index case.

**Table 1 biomedicines-10-00160-t001:** Clinical and demographic characteristics of *GBA* p.R202X heterozygote carriers.

Study	Diagnosis	Sex	Age-at-Onset	Age-at-Death or Last Assessed Normal	Autopsy-Confirmed	*APOE* Genotype	Ancestry
Present—Family 1	LBD	M	70	74	Y	3/3	Irish-American
Present—Family 1	LBSD	M	58	-	-	3/4	Irish-American
Present—Family 2	PD	F	36	-	-	3/3	British-American
Present—Family 2	Unaffected	F	-	72	-	-	British-American
Irwin, D.J., et al. [1]	LBSD	M	50	57	Y	3/3	NA
Mata, I.F., et al.[16], Sidransky, E., et al. [17], Chahine, L.M., et al. [18], Davis, M.Y., et al. [19]	PD	F	47	-	-	-	African-American
Barrett, M.J., et al. [20]	-	-	-	-	-	-	NA

M = Male; F = Female; Y = Yes; LBD = Lewy body dementia; LBSD = Lewy body spectrum disorder; NA = not available; PD = Parkinson’s disease.

## Data Availability

The data presented in this study are available on request from the corresponding author. The data are not publicly available due to privacy policies and Terms and Conditions agreements.

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
