# Peer review of "Phenotypic Heterogeneity among GBA p.R202X Carriers in Lewy Body Spectrum Disorders"

_biomedicines, 2022, doi:10.3390/biomedicines10010160_

Round 1

Reviewer 1 Report

Napolioni et al described clinical-genetic features of patients with Lewy Body Spec- 14 trum Disorder. The manuscript covers several aspects of the subject and is well organized. However, for a case report, the article is too long and needs to be reduced.

In addition, the authors need to pay attention to the presentation of figures, particularly figure 2 and 3.

Regarding the heterogeneity, the authors should be discussed the importance of  additional characterizations of these families (cytogenetics; telomere, DNA repair….)

Author Response

Napolioni et al described clinical-genetic features of patients with Lewy Body Spectrum Disorder. The manuscript covers several aspects of the subject and is well organized. However, for a case report, the article is too long and needs to be reduced.

RESPONSE: We thank the Reviewer for the positive evaluation. We recognize that the manuscript is a bit long to be considered a “Case report”. However, given the reviewer’s comment on the importance of additional characterization of these families, coupled with the need to have a detailed clinical/genomic characterization of subjects carrying the extremely rare GBA p.R202X mutation, we decided to change the categorization of our paper, from “Case report” to “Article”.

In addition, the authors need to pay attention to the presentation of figures, particularly figure 2 and 3.

RESPONSE: We thank the Reviewer for pointing out this problem. Indeed, we realized that the legends to figure 2 and 4 (not 3), were misleading. For both legends, the “affected son of the index case”, was mistakenly reported as “index case”. We fixed this inconsistency, accordingly.

Regarding the heterogeneity, the authors should be discussed the importance of additional characterizations of these families (cytogenetics; telomere, DNA repair….).

RESPONSE: We now acknowledge the other possible sources of biological interference with the GBA p.R202X related-phenotype expressivity. We stated in the discussion that “Nevertheless, we cannot exclude the possibility of other biological factors, not detectable by targeted NGS, as the basis of the phenotypic heterogeneity characterizing the affected members of the two families described here. Indeed, it has been shown that telomere length may influence the occurrence of dementia in PD patients [22]. Also, several studies have argued for the existence of genetic modifiers of PD clinical presentation, such as TMEM106B [23], SCNA [24], and COMT [25].

In the discussion, we also reported that “Undoubtedly, a deeper biological characterization of the two families described here, by performing a wider genomic investigation (e.g., by Whole-Genome Sequencing, Structural Variation typing), eventually coupled with the analysis of epigenomic changes (e.g., telomere length, methylation profiling), would help to identify new modifiers of clinical phenotypes in LBSD.”

Reviewer 2 Report

The authors, Napolioni et al., described an article in which they describe the clinical and neuropathologic features of patients with Lewy Body Spectrum Disorder (LBSD) carrying a nonsense variant, c.604C>T; p.R202X, in the Glucocerebrosidase 1 (GBA) gene.

The presence of the GBA p.R202X gene mutation in patients with LBSD and Parkinson’s disease (PD) has already previously published as correctly described in Table 1. It is a known information. Furthermore, data of important clinical interest are not reported. Thus, this manuscript does not contain any data of significant scientific curiosity and the interest to the readers is low.

Author Response

The presence of the GBA p.R202X gene mutation in patients with LBSD and Parkinson’s disease (PD) has already previously published as correctly described in Table 1. It is a known information. Furthermore, data of important clinical interest are not reported. Thus, this manuscript does not contain any data of significant scientific curiosity and the interest to the readers is low.

RESPONSE: The variant was already found and reported in the supplementary material of a previous work, but no clinical/detailed information was made available to interpret its clinical relevance. We feel our manuscript provides clinical geneticists with relevant information to confirm the pathogenicity, though with incomplete penetrance, of this extremely rare GBA variant.

Reviewer 3 Report

The purpose of the study submitted by Napolioni and colleagues was to describe clinical and neuropathological features in families with Lewy Body spectrum Disorder. They found a mutation of GBA gene associated to a phenotypic heterogeneity.

This is a well-planned study, with appropriate analyses. The manuscript is generally well written and the objective is clear.

Major point

I suggest to compare these cases with other LBSD cases described in literature that have mutations in GBA different from p.R202X or in other genes, such as PSENs

Author Response

I suggest to compare these cases with other LBSD cases described in literature that have mutations in GBA different from p.R202X or in other genes, such as PSENs

RESPONSE: We thank the reviewer for this constructive comment. However, we think that this request cannot be adequately addressed since hundreds of GBA variants have been already described in Lewy-Body Spectrum Disorders (without considering the number of variants in other genes, such as the PSENs, previously linked to LBSD). With our paper, we aim to describe a very rare, nonsense GBA variant that was previously reported only in one PD patient [16-19], one case with neuropathologically-verified LBSD [1] and in one unaffected relative of a Gaucher’s disease patient [20]. In particular, we want to highlight the fact that the additional subjects carrying this rare, nonsense GBA variant were identified through the extensive literature search we performed on more than 800 articles.

Round 2

Reviewer 2 Report

I confirm that it is a determining factor in the interest of the study that the mutation has already been published.

Author Response

RESPONSE: We have clearly acknowledged in the manuscript that the mutation has already been published. Indeed, a relevant section of our manuscript reports a comprehensive literature search of GBA p.R202X variant carriers. This information is novel and will help clinical geneticists to interpret the causal role of this variant.

As a matter of fact, there is no available literature reporting a genotype-phenotype description of this extremely rare, nonsense GBA variant. Published reports [refs 1, 16-19, 20] identified this variant in different groups of subjects (with LBSD, PD and Gaucher’s Disease, respectively), generally categorized as GBA mutation carriers, but no detailed description was ever provided. Most of the published reports on GBA genetic variability and LBD/PD/Gaucher’s Disease, have focused on the effect of a handful of relatively common GBA mutations (e.g., GBA p.N370S and p.L444P).

More specifically, the p.R202X variant is a nonsense mutation, whereas the majority of GBA mutations implicated in PD risk are missense mutations. There is a debate in the literature if GBA variants lead to a loss of function or toxic gain of function (https://onlinelibrary.wiley.com/doi/full/10.1111/ene.13837). Therefore, it is critical to also report cases in detail that carry mutations other than the common, recurrent missense mutations to understand the full spectrum of disease, decipher disease mechanism, and counsel patients about risk and disease trajectory.

In the clinical genetics’ community, we highly recognize the effort of describing and characterizing rare mutations, even if not completely novel, through genotype-phenotype analyses. Indeed, a very recent paper published in this journal, reported the genotype-phenotype correlation for a known mutation in the ACTB gene, causing Becker’s Nevus (Dai S, Wang H, Lin Z. ACTB Mutations Analysis and Genotype-Phenotype Correlation in Becker's Nevus. Biomedicines. 2021 Dec 10;9(12):1879. doi: 10.3390/biomedicines9121879.).

Given the biochemical nature (nonsense), and the lack of a comprehensive evaluation of GBA p.R202X in the literature, we feel our manuscript will be helpful both to medical geneticists counselling patients and to basic scientists in elucidating pathogenic mechanisms. Overall, we feel that the amount of novel/structured information delivered to the academic/clinical community by the present article will justify its publication in Biomedicines.

Round 3

Reviewer 2 Report

Although already commented, I confirm that the manuscript could be of interest to a journal with a lower impact factor since the mutation has already been published